# Trajectory-based Explainability Framework for Offline RL

**Shripad Deshmukh**[*,1]**, Arpan Dasgupta**[‡,2]**, Chirag Agarwal**[1]**, Nan Jiang**[3]**, Balaji Krishnamurthy**[1]**, Georgios Theocharous**[4]**, Jayakumar Subramanian**[1]

[1]Media and Data Science Research, Adobe, Noida, India.
[2]International Institute of Information Technology, Hyderabad, India.
[3]University of Illinois Urbana-Champaign, Champaign, United States.
[4]Adobe Research, San Jose, United States.

## Abstract

Explanation is a key component for the adoption of reinforcement learning (RL) in many real-world decision-making problems. In the literature, the explanation is often provided by saliency attribution to the features of the RL agent's state. In this work, we propose a complementary approach to these explanations, particularly for offline RL, where we attribute the policy decisions of a trained RL agent to the trajectories encountered by it during training. To do so, we encode trajectories in offline training data individually as well as collectively (encoding a set of trajectories). We then attribute policy decisions to a set of trajectories in this encoded space by estimating the sensitivity of the decision with respect to that set. Further, we demonstrate the effectiveness of the proposed approach in terms of quality of attributions as well as practical scalability in diverse environments that involve both discrete and continuous state and action spaces such as grid-worlds, video games (Atari) and continuous control (MuJoCo).

## 1  Introduction

Reinforcement learning (Sutton & Barto, 2018) has enjoyed great popularity and has achieved huge success, especially in the online settings, post advent of deep reinforcement learning (Mnih et al., 2013; Schulman et al., 2017; Silver et al., 2017; Haarnoja et al., 2018). Deep RL algorithms are now able to handle high-dimensional observations such as visual inputs with ease. However, using these algorithms in the real world requires i) efficient learning from minimal exploration to avoid catastrophic decisions due to insufficient knowledge of the environment, and ii) being explainable. The first aspect is extensively being studied under offline RL (Levine et al., 2020) where the agent is trained on collected experience rather than exploring directly in the environment (Kumar et al., 2020; Yu et al., 2020; Kostrikov et al., 2021). However, more work is needed to address the explainability aspect of RL decision-making.

Previously, researchers have attempted explaining decisions of RL agent by highlighting important features of the agent's state (input observation) (Puri et al., 2019; Iyer et al., 2018; Greydanus et al., 2018). While these approaches are useful, we take a complementary route. Instead of identifying salient state-features, we wish to identify the past experiences (trajectories) that led the RL agent to learn certain behaviours. We call this approach as trajectory-aware RL explainability. Such explainability confers faith in the decisions suggested by the RL agent in critical scenarios (surgical (Loftus et al., 2020), nuclear (Boehnlein et al., 2022), etc.) by looking at the trajectories

---

[*]Email for correspondence: `shdeshmu@adobe.com`

[‡] Work done during the Adobe MDSR Research Internship Program.

Offline Reinforcement Learning Workshop at Neural Information Processing Systems, 2022

responsible for the decision. While this sort of training data attribution has been shown to be highly effective in supervised learning (Nguyen et al., 2021), to the best of our knowledge, this is the first work to study data attribution-based explainability in RL. In the present work, we restrict ourselves to the offline RL setting.

Contributions of this work are enumerated below:

1. A novel explainability framework for reinforcement learning that aims to find experiences (trajectories) that lead an RL agent to learn certain behaviour.
2. A solution for trajectory attribution in offline RL setting based on state-of-the-art sequence modeling techniques. In our solution, we present a methodology that generates a single embedding for a trajectory of states, actions, and rewards. We also extend this method to generate a single encoding of data containing a set of trajectories.
3. Analysis of trajectory explanations produced by our technique along with analysis of the trajectory embeddings generated, where we demonstrate how different embedding clusters represent different semantically meaningful behaviours (see Appendix A.6).

## 2 Background and Related Work

**Explainability in RL.** Explainable AI (XAI) refers to the field of machine learning (ML) that focuses on developing tools for explaining the decisions of ML models. Explainable RL (XRL) (Puiutta & Veith, 2020) is a sub-field of XAI that specializes in interpreting behaviours of RL agents. Prior works include approaches that distill the RL policy into simpler models such as decision tree (Coppens et al., 2019) or to human understandable high-level decision language (Verma et al., 2018). However, such policy simplification fails to approximate the behavior of complex RL models. In addition, causality-based approaches (Pawlowski et al., 2020; Madumal et al., 2020) aim to explain an action's agent by identifying the cause behind it using counterfactual samples. To the best of our knowledge, for the first time, we explore the direction of explaining an agent's behaviour by attributing its actions to past encountered trajectories rather than highlighting state features.

**Sequence Modeling in Offline RL.** Recently, the RL problem of maximizing long-term return has been cast as taking the best possible action given the sequence of past interactions in terms of states, actions, rewards (Chen et al., 2021; Janner et al., 2021; Reed et al., 2022; Park et al., 2018). Such sequence modelling approaches to RL, especially the ones based on transformer architecture (Vaswani et al., 2017), have produced state-of-the-art results in various offline RL benchmarks, and offer rich latent representations to work with. However, little to no work has been done in the direction of understanding these sequence representations and their applications. In this work, we base our solution on these sequence modelling approaches to leverage their high efficiency in capturing the policy and environment dynamics of the offline RL systems.

## 3 Trajectory Attribution

**Preliminaries.** We denote the offline RL dataset using $\mathcal{D}$ that comprises a set of $n_\tau$ trajectories. Each trajectory $\tau_j \in \mathcal{D}$ comprises of a sequence of observation ($o_k$), action ($a_k$) and per-step reward ($r_k$) tuples with $k$ ranging from 1 to the length of the trajectory $\tau_j$. We begin by training an offline RL agent on this data using any standard offline RL algorithm.

**Algorithm.** Having obtained the learned policy using an offline RL algorithm, our objective now is to attribute this policy, i.e., the action chosen by this policy, at a given state to a set of trajectories. We intend to achieve this in the following way. We want to find the smallest set of trajectories, the absence of which from the training data leads to different behaviour at the state under consideration. That is, we posit that this set of trajectories contains specific behaviours and respective feedbacks from the environment that train the RL agent to make decisions in a certain manner. This identified set of trajectories would then be provided as attribution for the original decision.

While this basic idea is straightforward and intuitive, it is not computationally feasible beyond a few small RL problems with discrete state and action spaces. The key requirement to scale this approach to large, continuous state and action space problems, is to group the trajectories into clusters which can then be used to analyse their role in the decision-making of the RL agent. In this work, we propose to cluster the trajectories using trajectory embeddings produced with the help of state-of-the-art

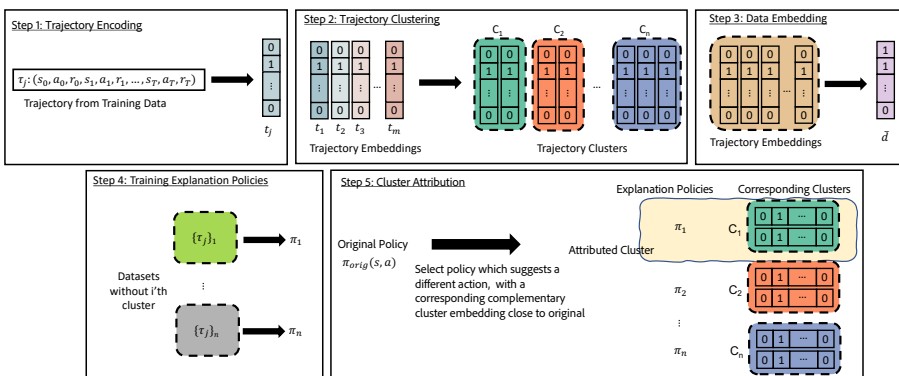

Figure 1: **Trajectory Attribution in Offline RL.** First, we encode trajectories in offline data using sequence encoders and then cluster the trajectories using these encodings. Also, we generate a single embedding for the data. Next, we train explanation policies on variants of the original dataset and compute corresponding data embeddings. Finally, we attribute decisions of RL agents trained on entire data to trajectory clusters using action and data embedding distances.

sequence modelling approaches. Fig. 1 gives an overview of our proposed approach, the steps of which are described below:

**(i) Trajectory Encoding.** First, we tokenize the trajectories in the offline data according to the specifications of the sequence encoder used (e.g. decision transformer/trajectory transformer). The observation, action and reward tokens of a trajectory are then fed to the sequence encoder to produce corresponding latent representations, which we refer to as output tokens. We define the *trajectory embedding* as an average of these output tokens. This technique is inspired by average-pooling techniques (Choi et al., 2021; Briggs, 2021) in NLP used to create sentence embedding from embeddings of words present in it. (Refer to Alg. 1.)

**(ii) Trajectory Clustering.** Having obtained trajectory embeddings, we cluster them using X-Means clustering algorithm (Pelleg et al., 2000) with implementation provided by Novikov (2019). While in principle, any suitable clustering algorithm can be used here, we chose X-Means as it is a simple extension to the K-means clustering algorithm (Lloyd, 1982); it determines the number of clusters $n_c$ automatically. This enables us to identify all possible patterns in the trajectories without forcing $n_c$ as a hyperparameter (Refer to Alg. 2).

**(iii) Data Embedding.** We need a way to identify the least change in the original data that leads to the change in behaviour of the RL agent. To achieve this, we propose a representation for data comprising collection of trajectories. The representation has to be agnostic to the order in which trajectories are present in the collection. So, we follow the set-encoding procedure prescribed in (Zaheer et al., 2017) where we first sum the embeddings of the trajectories in the collection, normalize this sum by division with a constant and further apply a non-linearity, in our case, simply, softmax over the feature dimension to generate a single *data embedding* (Refer to Alg. 3).

We use this technique to generate data embeddings for $n_c + 1$ sets of trajectories. The first set represents the entire training data whose embedding is denoted by $\bar{d}_{\text{orig}}$. The remaining $n_c$ sets are constructed as follows. For each trajectory cluster $c_j$, we construct a set with the entire training data but the trajectories contained in $c_j$. We call this set the complementary data set corresponding to cluster $c_j$ and the corresponding data embedding as the complementary data embedding $\bar{d}_j$.

**(iv) Training Explanation Policies.** In this step, for each cluster $c_j$, using its complementary data set, we train an offline RL agent. We ensure that all the training conditions (algorithm, weight initialization, optimizers, hyperparameters, etc.) are identical to the training of original RL policy, except for the modification in the training data. We call this newly learned policy as the explanation policy corresponding to cluster $c_j$. We thus get $n_c$ explanation policies at the end of this step. In addition, we compute data embeddings for complementary data sets (Refer to Alg. 4).

**(v) Cluster Attribution.** In this final step, given a state, we note the actions suggested by all the explanation policies at this state. We then compute the distances of these actions (where we

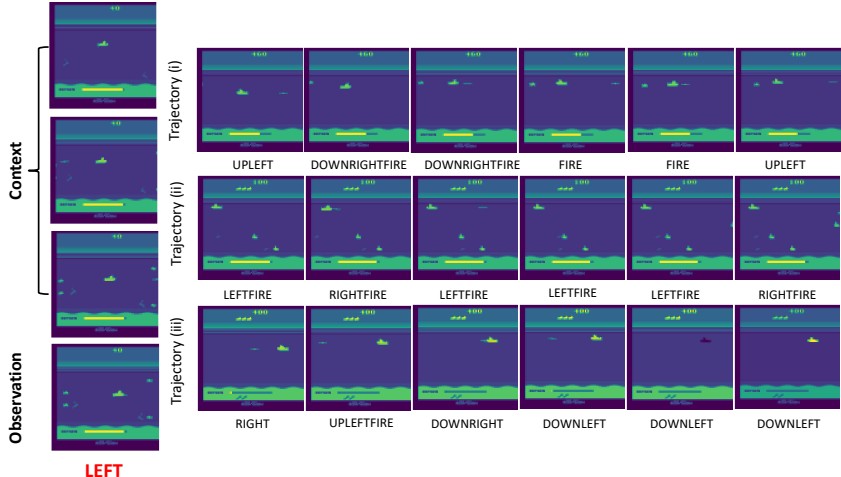

Figure 2: **Seaquest Trajectory Attribution.** The agent (submarine) decides to take 'left' for the given observation under the context provided. Top-3 attributed trajectories are shown on the right (for each traj., we show 6 sampled observations along with actions taken by behaviour policy). From these trajectories, we can infer how the agent is aligning itself to face enemies coming from left.

assume a metric over the action space) from the action suggested by the original RL agent at the state. The explanation policies corresponding to the maximum of these distances form the candidate attribution set. For each policy in this candidate attribution set, we compute the distance between its respective complementary data embedding and the data embedding of the entire training data using the Wasserstein metric for capturing distances between softmax simplices (Vallender, 1974). We then select the policy that has the smallest data distance and attribute the decision of the RL agent to the cluster corresponding to this policy(Refer to Alg. 5). Our approach comprised of all the five steps is summarized in Algorithm 6.

## 4 Results

Next, we present experimental results on the Atari Seaquest environment (Bellemare et al., 2013) to show the effectiveness of our approach in generating trajectory explanations. The details of the experimental setup are given in Sec. A.2.

Fig. 2 shows trajectory attribution results for RL agent playing Seaquest Atari game. Moreover, table 1 presents quantitative analysis of the proposed trajectory attribution using evaluation metrics described in Sec. A.2. Initial state value estimate ($\mathbb{E}(V(s_0))$) for original policy $\pi_{\text{orig}}$ compares with estimates for explanation policies trained on different complementary datasets. This indicates that $\pi_{\text{orig}}$, having access to all behaviours, is able to outperform other policies that are trained on data lacking information about important behaviours (e.g. fighting in the top-right corner). Further, local mean absolute action-value difference ($\mathbb{E}(|\Delta Q_{\pi_{\text{orig}}}|) = \mathbb{E}(|Q_{\pi_{\text{orig}}}(\pi_{\text{orig}}(s)) - Q_{\pi_{\text{orig}}}(\pi_j(s))|)$) and action difference ($\mathbb{E}(\mathbb{1}(\pi_{\text{orig}}(s) \neq \pi_j(s)))$) turn out to be highly correlated, i.e., the explanation policies that suggest the most contrasting actions are usually perceived by the original policy as low-return actions. This evidence supports the proposed trajectory algorithm as we want to identify the behaviours which when removed make agent choose actions that are not considered suitable originally. In addition, in columns 4 and 5, we provide normalized Wasserstein distances between the data embeddings ($W_{\text{dist}}(\bar{d}, \bar{d}_j)$) and the cluster attribution distribution ($\mathbb{P}(c_{\text{final}} = c_j)$), respectively. The latter depicts how RL decisions are dependent on various behaviour clusters.

The results on Grid-world and MuJoCo HalfCheetah (Todorov et al., 2012) are given in Sec. A.5.1; additional qualitative results are given in Sec. A.5.2; clustering analysis is provided in Sec. A.6.

Table 1: Quantitative Analysis of Seaquest Trajectory Attribution.

| $\pi$ | $\mathbb{E}(V(s_0))$ | $\mathbb{E}(|\Delta Q_{\pi_{\mathrm{orig}}}|)$ | $\mathbb{E}(\mathbb{1}(\pi_{\mathrm{orig}}(s) \neq \pi_j(s))$ | $W_{\mathrm{dist}}(\bar{d}, \bar{d}_j)$ | $\mathbb{P}(c_{\mathrm{final}} = c_j)$ |
|------|------|------|------|------|------|
| orig | 85.9977 | - | - | - | - |
| 0 | 50.9399 | 1.5839 | 0.9249 | 0.4765 | 0.1129 |
| 1 | 57.5608 | 1.6352 | 0.8976 | 0.9513 | 0.0484 |
| 2 | 66.7369 | 1.5786 | 0.9233 | **1.0000** | 0.0403 |
| 3 | 3.0056 | **1.9439** | **0.9395** | 0.8999 | 0.0323 |
| 4 | 58.1854 | 1.5813 | 0.8992 | 0.5532 | 0.0968 |
| 5 | 87.3034 | 1.6026 | 0.9254 | 0.2011 | **0.3145** |
| 6 | 70.8994 | 1.5501 | 0.9238 | 0.6952 | 0.0968 |
| 7 | **89.1832** | 1.5628 | 0.9249 | 0.3090 | 0.2581 |

## 5  Discussion

In this work, we proposed a novel explanation technique that attributes decisions suggested by an RL agent to trajectories encountered by the agent in the past. We provided an algorithm that enables us to perform trajectory attribution in offline RL. The key idea behind our approach was to study sensitivity of original RL agent's policy to the trajectory clusters obtained using sequence modelling techniques. We demonstrated the utility of our method using experiments in grid-world, Seaquest and HalfCheetah environments.

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

## A   Appendix

### A.1   Algorithms

---

**Algorithm 1:** encodeTrajectories

---

/* Encoding given set of trajectories individually                          */

**Input**     :Offline Data $\{\tau_i\}$, Sequence Encoder $E$

**Initialize :** Initialize array $T$ to collect the trajectory embeddings

1 **for** $\tau_j$ *in* $\{\tau_i\}$ **do**

    /* Using $E$, get output tokens for all the $o$, $a$ & $r$ in $\tau_j$          */

2    $(e_{o_{1,j}}, e_{a_{1,j}}, e_{r_{1,j}}, ..., e_{o_{\mathbb{T},j}}, e_{a_{\mathbb{T},j}}, e_{r_{\mathbb{T},j}}) \leftarrow E(o_{1,j}, a_{1,j}, r_{1,j}, ..., o_{\mathbb{T},j}, a_{\mathbb{T},j}, r_{\mathbb{T},j})$ // where $3\mathbb{T}$ = #input tokens

    /* Take mean of outputs to generate $\tau_j$'s embedding $t_j$          */

3    $t_j \leftarrow (e_{o_{1,j}} + e_{a_{1,j}} + e_{r_{1,j}} + e_{o_{2,j}} + e_{a_{2,j}} + e_{r_{2,j}} + ... + e_{o_{\mathbb{T},j}} + e_{a_{\mathbb{T},j}} + e_{r_{\mathbb{T},j}})/(3\mathbb{T})$

4    Append $t_j$ to $T$

**Output**   :Return the trajectory embeddings $T = \{t_i\}$

---

---

**Algorithm 2:** clusterTrajectories

---

```
/* Clustering the trajectories using their embeddings          */
```
**Input**   : Trajectory embeddings $T = \{t_i\}$, clusteringAlgo
1 $C \leftarrow$ clusteringAlgo(T) `// Cluster using provided clustering algorithm`
**Output**   : Return trajectory clusters $C = \{c_i\}_{i=1}^{n_c}$

---

---

**Algorithm 3:** generateDataEmbedding

---

```
/* Generating data embedding for a given set of trajectories   */
```
**Input**   : Trajectory embeddings $T = \{t_i\}$, Normalizing factor $M$, Softmax temperature $T_{\text{soft}}$
1 $\bar{s} \leftarrow \frac{\sum_i t_i}{M}$ `// Sum the trajectory embeddings and normalize them`
2 $\bar{d} \leftarrow \{d_j | d_j = \frac{exp(s_j/T_{\text{soft}})}{\sum_k exp(s_k/T_{\text{soft}})}\}$ `// Take softmax along feature dimension`
**Output**   : Return the data embedding $\bar{d}$

---

---

**Algorithm 4:** trainExpPolicies

---

```
/* Train explanation policies and compute corresponding data embeddings
   */
```
**Input**   : Offline data$\{\tau_i\}$, Traj. Embeddings $T$, Traj. Clusters $C$, offlineRLAlgo
1 **for** $c_j$ *in* $C$ **do**
2 $\quad$ $\{\tau_i\}_j \leftarrow \{\tau_i\} - c_j$ `// Compute complementary dataset corresp. to` $c_j$
3 $\quad$ $T_j \leftarrow$ gatherTrajectoryEmbeddings$(T, \{\tau_i\}_j)$ `// Gather corresp.` $\tau$ `embeds`
4 $\quad$ Explanation policy, $\pi_j \leftarrow$ offlineRLAlgo$(\{\tau_i\}_j)$
5 $\quad$ Complementary data embedding, $\bar{d}_j \leftarrow$ generateDataEmbedding$(T_j, M, T_{\text{soft}})$

**Output**   : Explanation policies $\{\pi_j\}$, Complementary data embeddings $\{\bar{d}_j\}$

---

---

**Algorithm 5:** generateClusterAttribution

---

```
/* Generating cluster attributions for aorig = πorig(s)          */
```
**Input**   : State $s$, Original Policy $\pi_{\text{orig}}$, Explanation Policies $\{\pi_j\}$, Original Data Embedding
$\quad\quad\quad$ $\bar{d}_{\text{orig}}$, Complementary Data Embeddings $\{\bar{d}_j\}$
1 Original action, $a_{\text{orig}} \leftarrow \pi_{\text{orig}}(s)$
2 Actions suggested by explanation policies, $a_j \leftarrow \pi_j(s)$
3 $d_{a_{\text{orig}}, a_j} \leftarrow$ calcActionDistance$(a_{\text{orig}}, a_j)$`// Compute action distance`
4 $K \leftarrow$ argmax$(d_{a_{\text{orig}}, a_j})$`// Get candidate clusters using argmax`
5 $w_k \leftarrow W_{\text{dist}}(\bar{d}_{\text{orig}}, \bar{d}_k)$`// Compute Wasserstein distance b/w complementary data`
`    embeddings of candidate clusters & orig data embedding`
6 $c_{\text{final}} \leftarrow$ argmin$(w_k)$`// Choose cluster with min data embedding dist.`
**Output**   : $c_{\text{final}}$

---

**Algorithm 6:** Trajectory Attribution in Offline RL

---

**Input** : Offline Data $\{\tau_i\}$, States needing explanation $\mathcal{S}_{\text{exp}}$, Sequence Encoder $E$,
offlineRLAlgo, clusteringAlgo, Normalizing constant $M$, Softmax Temperature $T_{\text{soft}}$

```
/* Train original offline RL policy                                    */
```
1 $\pi_{\text{orig}} \leftarrow$ offlineRLAlgo($\{\tau_i\}$)
```
/* Encode individual trajectories                                      */
```
2 $T =$ encodeTrajectories($\{\tau_i\}, E$) // Algo. 1
```
/* Cluster the trajectories                                            */
```
3 $C \leftarrow$ clusterTrajectories(T, clusteringAlgo) // Algo. 2
```
/* Compute data embedding for the entire dataset                       */
```
4 $\bar{d}_{\text{orig}} =$ generateDataEmbedding(T, $M, T_{\text{soft}}$) // Algo. 3
```
/* Generate explanation policies and their corresponding complementary
   data embeddings                                                     */
```
5 $\{\pi_j\}, \{\bar{d}_j\} \leftarrow$ trainExpPolicies($\{\tau_i\}$, T, $C$, offlineRLAlgo)// Algo. 4
```
/* Attributing policy decisions for given set of states               */
```
6 **for** $s \in \mathcal{S}_{\text{exp}}$ **do**
7    $c_{\text{final}} \leftarrow$ generateClusterAttribution($s, \pi_{\text{orig}}, \{\pi_j\}, \bar{d}_{\text{orig}}, \bar{d}_j$) // Algo. 5
8    *Optionally, select top N trajectories in the cluster $c_{\text{final}}$ using a pre-defined criteria.

---

## A.2 Experimental Setup

We first describe the environments, models, and metrics designed to study the reliability of our trajectory explanations.

**RL Environments.** We perform experiments on three environments: i) *Grid-world* (Fig. 3) which has discrete state and action spaces, ii) *Seaquest* from Atari suite which has environments with continuous visual observations and discrete action spaces (Bellemare et al., 2013), and iii) *HalfCheetah* from MuJoCo environments which are control environments with continuous state and action spaces (Todorov et al., 2012).

**Offline Data and Sequence Encoders.** For grid-world, we collect offline data of 60 trajectories from policy rollout of other RL agents and train an LSTM-based trajectory encoder following the procedure described in trajectory transformer, replacing the transformer with LSTM. For Seaquest, we collect offline data of 717 trajectories from the D4RL-Atari repository and use a pre-trained decision transformer as trajectory encoder. Similarly, for HalfCheetah, we collect offline data of 1000 trajectories from the D4RL repository (Fu et al., 2020) and use a pre-trained trajectory transformer as a trajectory encoder. To cluster high-level skills in long trajectory sequences, we divide the Seaquest trajectories into 30-length sub-trajectories and the HalfCheetah trajectories into 25-length sub-trajectories. These choices were made based on the transformers' input block sizes and the quality of clustering.

**Offline RL Training and Data Embedding.** We train the offline RL agents for each environment using the data collected as follows - for grid-world, we use model-based offline RL, and for Seaquest and HalfCheetah, we employ DiscreteSAC (Christodoulou, 2019) and SAC (Haarnoja et al., 2018), respectively, using d3rlpy implementations (Takuma Seno, 2021). We compute data embedding of entire training data for each of the environments. See Sec. A.4 for additional training details.

**Encoding of Trajectories and Clustering.** We encode the trajectory data using sequence encoders and cluster the output trajectory embeddings using the X-means algorithm. More specifically, we obtain 10 trajectory clusters for grid-world, 8 for Seaquest, and 10 for HalfCheetah.

**Complementary Data Sets.** We obtain complementary data sets using the aforementioned cluster information and provide 10 complementary data sets for grid-world, 8 for Seaquest, and 10 for HalfCheetah. Next, we compute data embeddings corresponding to these newly formed data sets.

---

https://github.com/takuseno/d4rl-atari
https://huggingface.co/edbeeching/decision_transformer_atari
https://github.com/Farama-Foundation/D4RL
https://github.com/jannerm/trajectory-transformer

**Explanation Policies.** Subsequently, we train explanation policies on the complementary data sets for each environment. The training produces 10 additional policies for grid-world, 8 policies for Seaquest, and 10 policies for HalfCheetah. In summary, we train the original policy on the entire data, obtain data embedding for the entire data, cluster the trajectories and obtain their explanation policies and complementary data embeddings.

**Trajectory Attribution.** Finally, we attribute a decision made by the original policy for a given state to a trajectory cluster. We choose top-3 trajectories from these attributed clusters by matching the context for the state-action under consideration with trajectories in the cluster in our experiments.

**Evaluation Metrics.** We compare policies trained on different data using three metrics (deterministic nature of policies is assumed throughout the discussion) – *1) Initial State Value Estimate* denoted by $\mathbb{E}(V(s_0))$ which is a measure of expected long-term returns to evaluate offline RL training as described in Paine et al. (2020), *2) Local Mean Absolute Action-Value Difference:* defined as $\mathbb{E}(|\Delta Q_{\pi_{\text{orig}}}|) = \mathbb{E}(|Q_{\pi_{\text{orig}}}(\pi_{\text{orig}}(s)) - Q_{\pi_{\text{orig}}}(\pi_j(s))|)$ that measures how original policy perceives suggestions given by explanation policies, and *3) Action Contrast Measure:* a measure of difference in actions suggested by explanation policies and the original action. Here, we use $\mathbb{E}(\mathbb{1}(\pi_{\text{orig}}(s) \neq \pi_j(s))$ for discrete action space and $\mathbb{E}((\pi_{\text{orig}}(s) - \pi_j(s))^2)$ for continuous action space. Further, we compute distances between embeddings of original and complementary data sets using Wasserstein metric: $W_{\text{dist}}(\bar{d}_{\text{orig}}, \bar{d}_j)$, later normalized to [0, 1]. Finally, the cluster attribution frequency is measured using metric $\mathbb{P}(c_{\text{final}} = c_j)$.

## A.3  Grid-world Environment Details

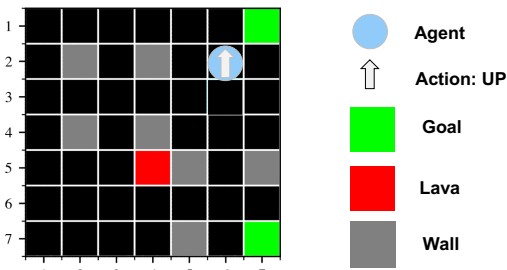

Figure 3: **Overview of the Grid-world Environment.** The aim of the agent is to reach any of the goal states (green squares) by avoiding lava (red square) and going around the impenetrable walls (grey squares). The reward for reaching the goal is +1; if the agent falls into the lava, it is -1. For any other transitions, the agent receives -0.1. The agent is allowed to take up, down, left or right as the action.

## A.4  Addtional Training Details

1. **Seaquest Atari Environment** – We employed Discrete SAC to train the original policy along with explanation policies, where the training was performed until saturation in the performance. We used the critic learning rate of $3 \times 10^{-4}$ and the actor learning rate of $3 \times 10^{-4}$ with a batch size of 256. The trainings were performed parallelly on a single Nvidia-A100 GPU hardware.

2. **HalfCheetah MuJoCo Environment** – We used SAC to train the original policy as well as explanation policies where we trained the agents until training performance saturated. We again used the critic learning rate of $3 \times 10^{-4}$ and the actor learning rate of $3 \times 10^{-4}$ with a batch size of 512. The policy trainings were performed parallelly on a single Nvidia-A100 GPU.

## A.5 Additional Trajectory Attribution Results

### A.5.1 Trajectory Attribution Results for Grid-world and HalfCheetah

**Qualitative Results.** Fig. 4 depicts a grid-world state - (1, 1), corresponding decision by the trained offline RL agent - 'right', and attribution trajectories explaining the decision. As we can observe, the decision is influenced not only by trajectory (traj.-i) that goes through (1, 1) but also by other distant trajectories(trajs.-ii, iii). These examples demonstrate that distant experiences (e.g. traj.-iii) could significantly influence the RL agent's decisions, deeming trajectory attribution an essential component of future XRL techniques.

Further, Fig. 5 shows HalfCheetah observation, the agent suggested in terms of hinge torques, and corresponding attributed trajectories showing runs that influence suggested set of torques. Additional results are given in Sec. A.5.

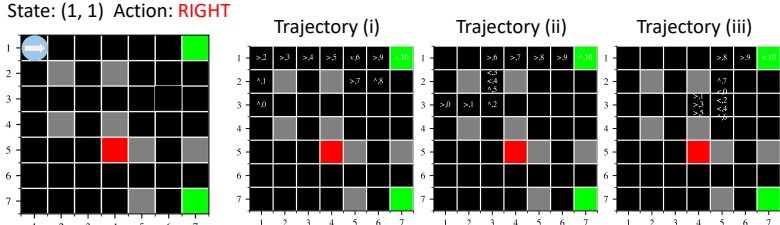

Figure 4: **Grid-world Trajectory Attribution.** RL agent suggests taking action 'right' in grid cell (1,1) which is attributed to trajectories (i), (ii) and (iii) (Here a trajectory is annotated by $\wedge,\vee,>,<$ arrows for 'up', 'down', 'right', 'left' actions respectively, along with the time-step (starting with 0) associated with the actions). The figure shows that the action suggestion is influenced by trajectories distant from the cell under consideration.

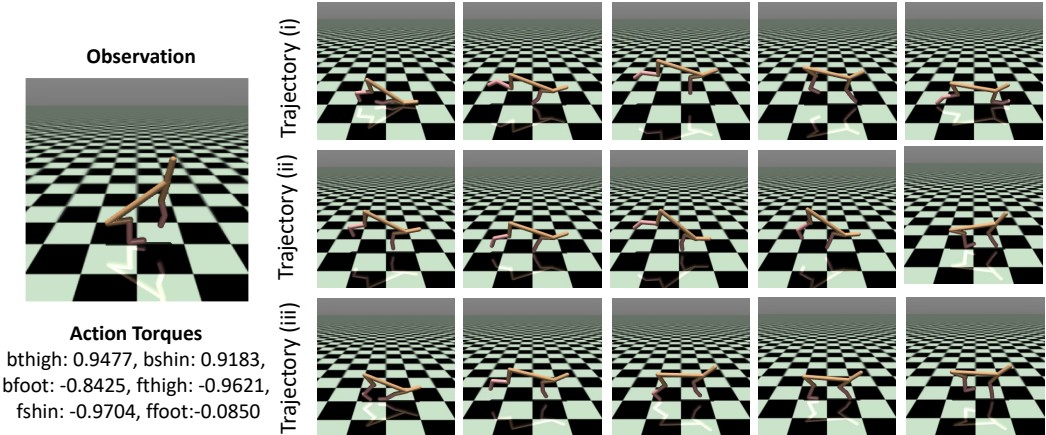

Figure 5: **HalfCheetah Trajectory Attribution.** The figure shows agent suggesting torques on different hinges for current position of the cheetah frame. The decision is influenced by the runs of cheetah shown on the right where the cheetah is getting up from the floor (here we show 5 sampled frames for a trajectory). Context is not shown here because unlike Seaquest environment, HalfCheetah decisions are made directly on a given observation (Puterman, 2014; Kaelbling et al., 1998).

**Quantitative Results.** Tables 2 and 3 present quantitative analysis of the proposed trajectory attribution for Grid-world and HalfCheetah respectively. The initial state value estimate for the original policy $\pi_{\text{orig}}$ matches or exceeds estimates for explanation policies trained on different complementary data sets in both environment settings. This indicates that the original policy, having access to all behaviours, is able to outperform other policies that are trained on data lacking information about important behaviours (e.g. grid-world: reaching a distant goal, HalfCheetah:

stabilizing the frame while taking strides). Furthermore, local mean absolute action-value difference and action differences turn out to be highly correlated (Tab. 1 and 3), i.e., the explanation policies that suggest the most contrasting actions are usually perceived by the original policy as low-return actions. This evidence supports the proposed trajectory algorithm as we want to identify the behaviours which when removed make agent choose actions that are not considered suitable originally. In addition, we provide the distances between the data embeddings in the penultimate column. The cluster attribution distribution is represented in the last column which depicts how RL decisions are dependent on various behaviour clusters. Interestingly, in the case of grid-world, we found that only a few clusters containing information about reaching goals and avoiding lava had the most significant effect on the original RL policy.

Table 2: **Quantitative Analysis of Grid-world Trajectory Attribution.** The analysis is provided using 5 metrics. Higher the $\mathbb{E}(V(s_0))$, better is the trained policy. High $\mathbb{E}(|\Delta Q_{\pi_{\text{orig}}}|))$ along with high $\mathbb{E}(\mathbb{1}(\pi_{\text{orig}}(s) \neq \pi_j(s))$ is desirable. The policies with lower $W_{\text{dist}}(\bar{d}, \bar{d}_j)$ and high action contrast are given priority while attribution. The cluster attribution distribution is given in the final column.

| $\pi$ | $\mathbb{E}(V(s_0))$ | $\mathbb{E}(|\Delta Q_{\pi_{\text{orig}}}|))$ | $\mathbb{E}(\mathbb{1}(\pi_{\text{orig}}(s) \neq \pi_j(s))$ | $W_{\text{dist}}(\bar{d}, \bar{d}_j)$ | $\mathbb{P}(c_{\text{final}} = c_j)$ |
|---|---|---|---|---|---|
| orig | **0.3061** | - | - | - | - |
| 0 | 0.3055 | 0.0012 | 0.0409 | **1.0000** | 0.0000 |
| 1 | 0.3053 | 0.0016 | 0.0409 | 0.0163 | 0.0000 |
| 2 | 0.3049 | 0.0289 | **0.1429** | 0.0034 | 0.0000 |
| 3 | 0.2857 | **0.0710** | 0.1021 | 0.0111 | 0.3750 |
| 4 | 0.2987 | 0.0322 | **0.1429** | 0.0042 | 0.1250 |
| 5 | 0.3057 | 0.0393 | 0.0409 | 0.0058 | 0.0000 |
| 6 | 0.3046 | 0.0203 | 0.1225 | 0.0005 | **0.5000** |
| 7 | 0.3055 | 0.0120 | 0.0205 | 0.0006 | 0.0000 |
| 8 | 0.3057 | 0.0008 | 0.0205 | 0.0026 | 0.0000 |
| 9 | 0.3046 | 0.0234 | **0.1429** | 0.1745 | 0.0000 |

Table 3: **Quantitative Analysis of HalfCheetah Trajectory Attribution.** The analysis is provided using 5 metrics. Higher the $\mathbb{E}(V(s_0))$, better is the trained policy. High $\mathbb{E}(|\Delta Q_{\pi_{\text{orig}}}|))$ along with high $\mathbb{E}((\pi_{\text{orig}}(s) - \pi_j(s))^2)$ is desirable. The policies with lower $W_{\text{dist}}(\bar{d}, \bar{d}_j)$ and high action contrast are given priority while attribution. The cluster attribution distribution is given in the final column.

| $\pi$ | Performance Metrics | | | | |
|---|---|---|---|---|---|
| | $\mathbb{E}(V(s_0))$ | $\mathbb{E}(|\Delta Q_{\pi_{\text{orig}}}|))$ | $\mathbb{E}((\pi_{\text{orig}}(s) - \pi_j(s))^2)$ | $W_{\text{dist}}(\bar{d}, \bar{d}_j)$ | $\mathbb{P}(c_{\text{final}} = c_j)$ |
| orig | 131.5449 | - | - | - | - |
| 0 | 127.1652 | 0.5667 | 0.6359 | 0.2822 | 0.0143 |
| 1 | 118.5663 | 0.4796 | 0.5633 | 0.0396 | 0.0214 |
| 2 | 122.0661 | 0.6904 | 0.9366 | 0.0396 | 0.1464 |
| 3 | 133.4590 | 0.5360 | 0.6611 | 0.0396 | 0.1250 |
| 4 | 118.3447 | 0.5622 | 0.6194 | **1.0000** | 0.0964 |
| 5 | **138.7517** | 0.6439 | 0.8262 | 0.0316 | 0.0893 |
| 6 | 120.7088 | 0.4740 | 0.4803 | 0.8813 | 0.0214 |
| 7 | 135.6848 | 0.5154 | 0.5489 | 0.0394 | **0.2036** |
| 8 | 113.3490 | 0.7826 | 1.0528 | 0.7067 | 0.1214 |
| 9 | 83.6211 | **0.9702** | **1.3453** | 0.0264 | 0.1607 |

### A.5.2 Additional Qualitative Results

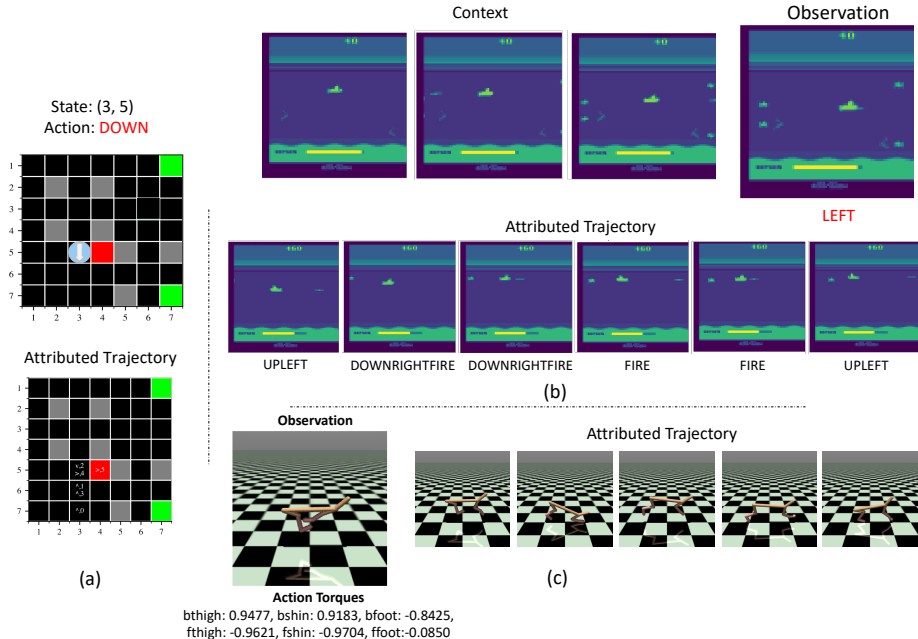

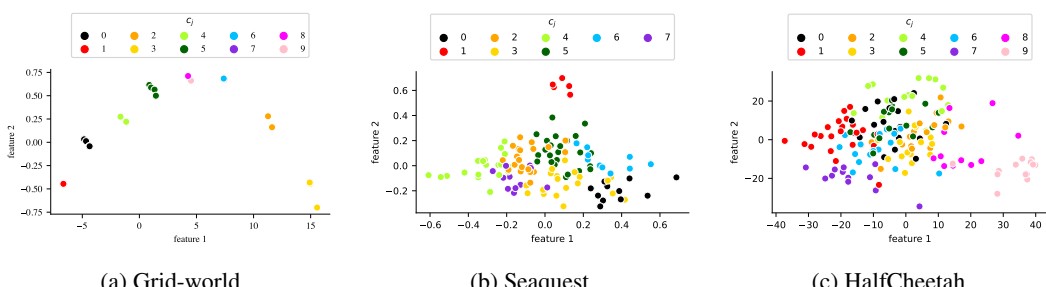

Figure 6: **Additional Trajectory Attribution Results.** Here we show randomly chosen trajectory from the top-3 attributed trajectories.a) Grid-world agent suggests taking 'DOWN' in cell (3,5) due to the attributed trajectory leading to lava. b) Seaquest agent suggests taking 'LEFT' and the corresponding attributed trajectory. c) HalfCheetah agent suggests a particular set of torques and the run found responsible for the same is shown side-by-side.

## A.6 Clustering Analysis

(a) Grid-world       (b) Seaquest       (c) HalfCheetah

Figure 7: **PCA Plot depicting Clusters of Trajectory Embeddings** for a) Grid-world, b) Seaquest, and c) HalfCheetah. We find that these clusters represent semantically meaningful high-level behaviours.

We observe that the trajectory embeddings obtained from the sequence encoders when clustered together 7 demonstrate characteristic high-level behaviours. For instance, in the case of grid-world (Refer 8), the clusters comprise semantically similar trajectories where the agent demonstrates behaviours such as '*falling into the lava*', '*achieving the goal in the first quadrant*', '*mid-grid journey to the goal*', etc. For Seaquest 9, we obtain trajectory clusters that represent high-level behaviours such as '*filling in oxygen*', '*fighting along the surface*', '*submarine bursting due to collision with enemy*', etc. and for HalfCheetah in Fig. 10, we obtain trajectory clusters that represent high-level actions such as '*taking long forward strides*', '*jumping on hind leg*', '*running with head down*', etc.

Although these results look quite promising, in this work we mainly focus on trajectory attribution that leverages these findings. In future, we wish to analyse the trajectory embeddings and the behaviour patterns in greater detail.

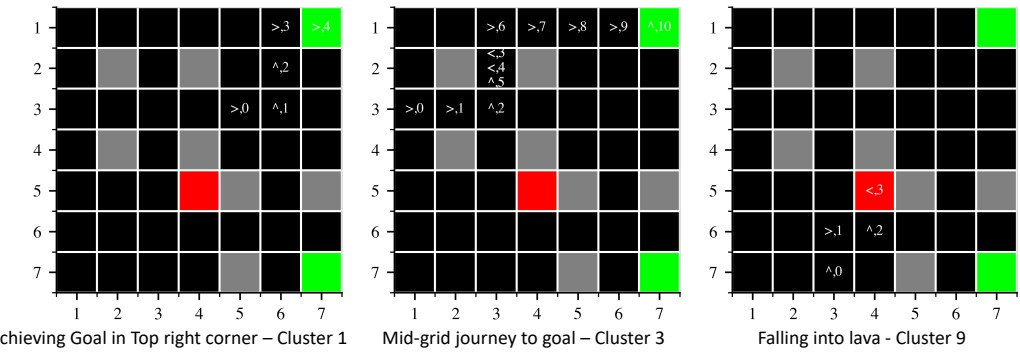

Figure 8: **Cluster Behaviours for Grid-world.** The figure shows 3 example high-level behaviours along with the action description and id of the cluster representing such behaviour.

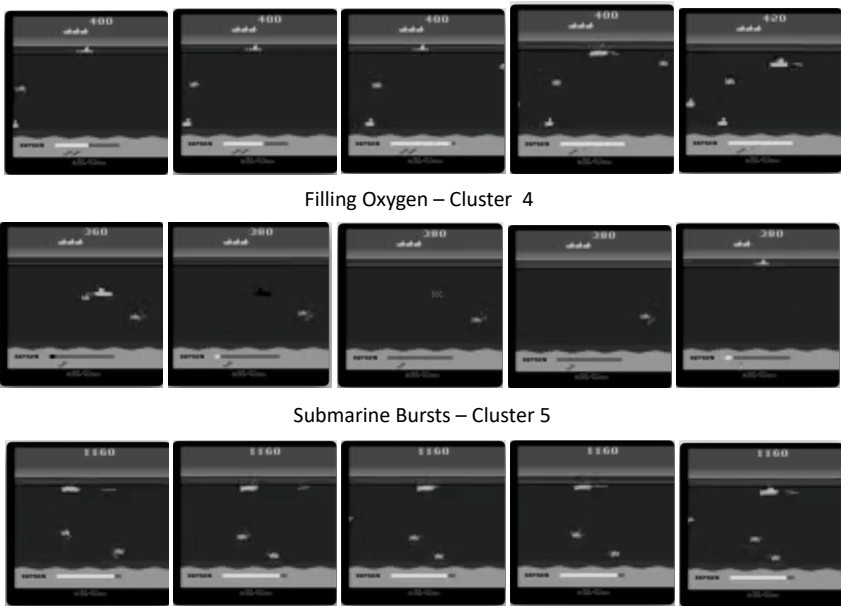

Filling Oxygen – Cluster 4

Submarine Bursts – Cluster 5

Fighting with Head Out – Cluster 7

Figure 9: **High-level Behaviours found in clusters for Seaquest** formed using trajectory embeddings produced using decision transformer. The figure shows 3 example high-level behaviours along with the action description and id of the cluster representing such behaviour.

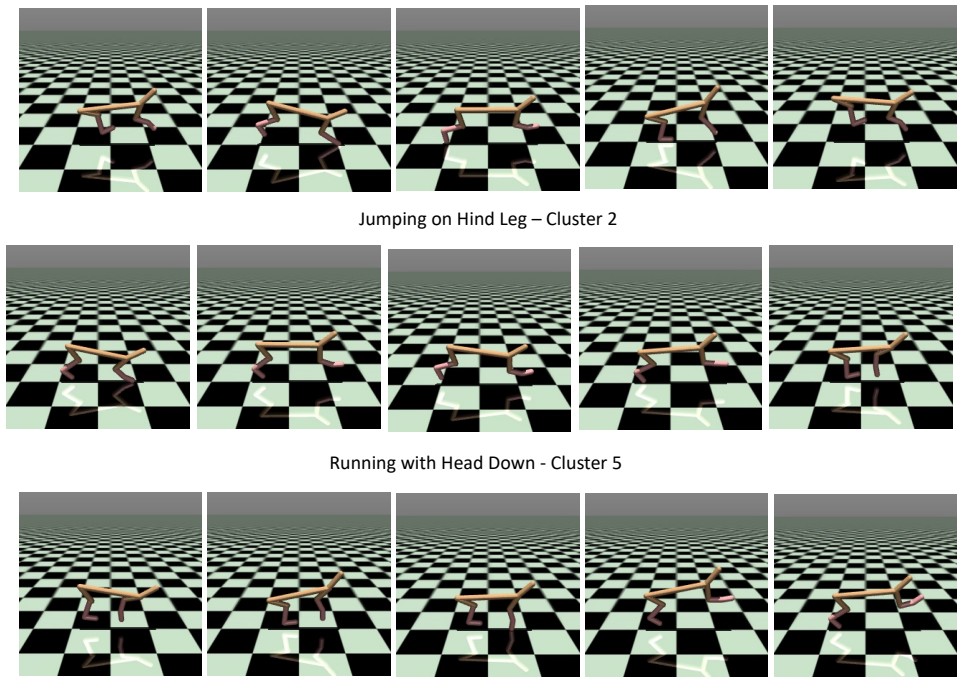

Jumping on Hind Leg – Cluster 2

Running with Head Down - Cluster 5

Forward Stride – Cluster 9

Figure 10: **High-level Behaviours found in clusters for HalfCheetah** formed using trajectory embeddings produced using trajectory transformer. The figure shows 3 example high-level behaviours along with the action description and id of the cluster representing such behaviour.

## A.7 Trajectory Attribution across algorithms

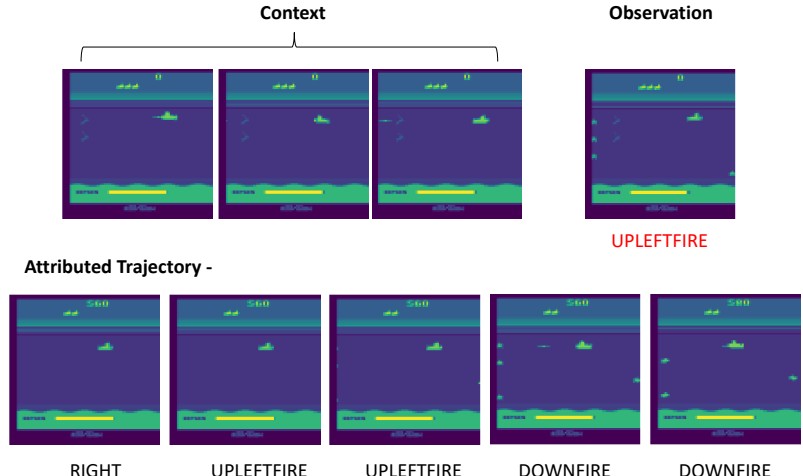

Figure 11: **Trajectory Attribution for DiscreteBCQ-trained Seaquest Agent.** The action 'UP-LEFTFIRE' is explained by our algorithm in terms of corresponding attributed trajectory. The action helps align agent to face enemies from left (akin to Fig. 2).

In Sec. A.5.1, we perform trajectory attribution on Seaquest environment trained using Discrete SAC algorithm. Here, we show results of our attribution algorithm in identifying influential trajectories for agents trained on same data but with different RL algorithm. Specifically, we choose Discrete Batch Constrained Q-Learning (**??**) to train a Seaquest policy.

Fig. 11 depicts a qualitative explanation generated using our algorithm for DiscreteBCQ trained agent. Table 4 gives quantitative numbers associated with attributions performed in this setting. It is quite interesting to note that our proposed algorithm assigns similar importance to various clusters as done in Table 1. That is, we find that certain behaviours in the data agnostic to the algorithm used for training play similar role in determining final execution policy. Thus, we find that our algorithm is generalizable and reliable enough to provide consistent insights across various RL algorithms.

Table 4: **Analysis of Trajectory Attribution for DiscreteBCQ-trained Seaquest Agent.**

| $\pi$ | $\mathbb{E}(V(s_0))$ | $\mathbb{E}(\lvert\Delta Q_{\pi_{\text{orig}}}\rvert)$ | $\mathbb{E}(\mathbb{1}(\pi_{\text{orig}}(s) \neq \pi_j(s))$ | $W_{\text{dist}}(\bar{d}, \bar{d}_j)$ | $\mathbb{P}(c_{\text{final}} = c_j)$ |
|---|---|---|---|---|---|
| orig | **1.3875** | - | - | - | - |
| 0 | 0.9619 | 0.1309 | 0.9249 | 0.4765 | 0.1025 |
| 1 | 0.5965 | **0.1380** | 0.8976 | 0.9513 | 0.0256 |
| 2 | 1.0157 | 0.1325 | 0.9233 | **1.0000** | 0.00854 |
| 3 | 1.1270 | 0.1323 | **0.9395** | 0.8999 | 0.0769 |
| 4 | 1.2243 | 0.1280 | 0.8992 | 0.5532 | 0.1025 |
| 5 | 1.2143 | 0.1367 | 0.9254 | 0.2011 | **0.3248** |
| 6 | 0.9752 | 0.1334 | 0.9238 | 0.6952 | 0.1196 |
| 7 | 1.1229 | 0.1352 | 0.9249 | 0.3090 | 0.2393 |

## A.8 Trajectory Attribution on Atari Breakout Environment

We present attribution results on additional environment of Atari Breakout Bellemare et al. (2013) trained using Discrete BCQ. Fig. 12 shows an instance of qualitative result and table 5 gives the number for overall attributions performed on Breakout.

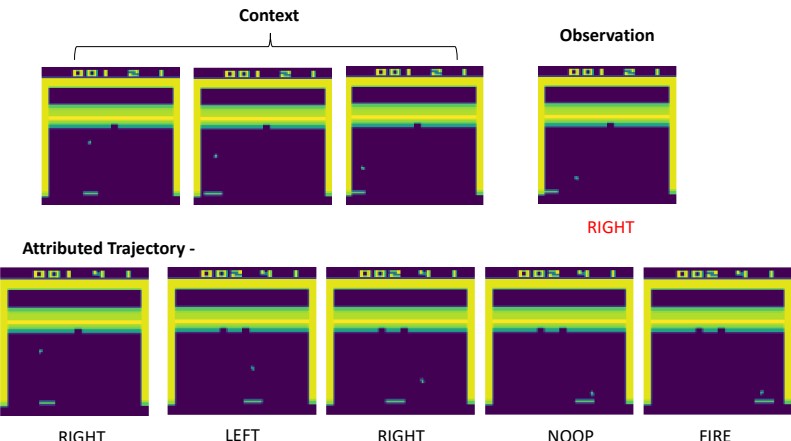

Figure 12: **Trajectory Attribution for DiscreteBCQ-trained Breakout Agent.** The agent proposes taking 'RIGHT' in the given observation frame. The corresponding attribution result shows how the ball coming from left would be played if moved to right.

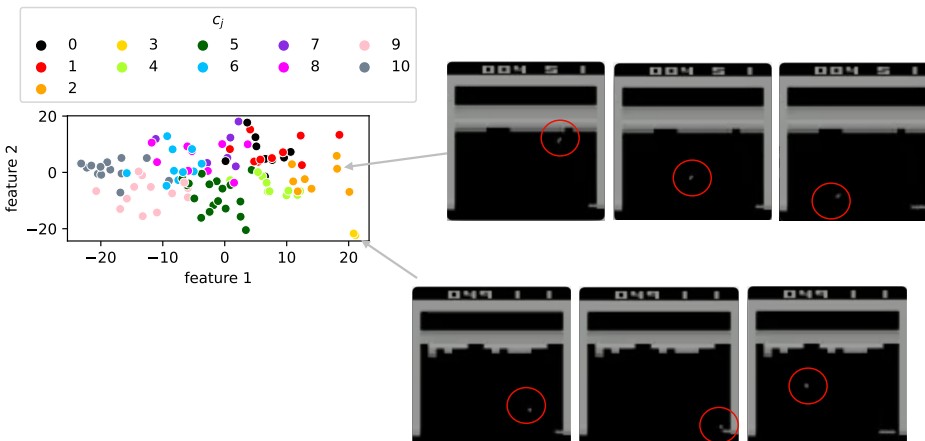

Figure 13: **Breakout Trajectory Clusters.** The figure shows PCA plot of breakout trajectories clustered into 11 clusters. We annotate cluster 2 showing ball hit from bottom right corner and cluster 3 where agent loses one life.

Table 5: **Quantitative Analysis of Trajectory Attribution for DiscreteBCQ-trained Breakout Agent.** We identify that clusters 2 and 3 representing 'corner shots from right' and 'depletion of a life' and impact the decision making significantly 13. This is insightful given how important these behaviours are in general, the first one showing how to avoid ending the game prematurely and the second one is well-known strategy in Breakout for playing at the end of right frame to break the walls on top left for creating a tunnel.

| $\pi$ | $\mathbb{E}(V(s_0))$ | $\mathbb{E}(|\Delta Q_{\pi_{\text{orig}}}|)$ | $\mathbb{E}(\mathbb{1}(\pi_{\text{orig}}(s) \neq \pi_j(s))$ | $W_{\text{dist}}(\bar{d}, \bar{d}_j)$ | $\mathbb{P}(c_{\text{final}} = c_j)$ |
|---|---|---|---|---|---|
| orig | 1.4570 | - | - | - | - |
| 0 | 1.1877 | 0.0972 | 0.7469 | 0.8828 | 0.0000 |
| 1 | **1.5057** | 0.0990 | 0.7317 | 0.2046 | 0.1428 |
| 2 | 1.2107 | 0.0983 | 0.7405 | 0.1676 | 0.2619 |
| 3 | 1.3946 | 0.0930 | 0.6687 | 0.1417 | **0.3095** |
| 4 | 1.4533 | 0.1043 | 0.7225 | 0.3827 | 0.0476 |
| 5 | 1.4678 | 0.1030 | 0.7310 | 0.5339 | 0.0000 |
| 6 | 1.1719 | 0.1022 | 0.7322 | **1.0000** | 0.0000 |
| 7 | 1.3493 | **0.1092** | 0.7225 | 0.3935 | 0.0000 |
| 8 | 1.2775 | 0.0916 | **0.7604** | 0.6999 | 0.04761 |
| 9 | 1.3773 | 0.0956 | 0.7496 | 0.5700 | 0.04761 |
| 10 | 1.4351 | 0.0998 | 0.7520 | 0.3005 | 0.1428 |

