# OpenReview forum: "Trajectory-based Explainability Framework for Offline RL"
_NeurIPS.cc/2022/Workshop/Offline_RL — Offline RL Workshop NeurIPS 2022_

### Official Review · Reviewer_Nsmh · 2022-10-11

**Rating:** 5
**Confidence:** 4

**Review:**

This paper proposes to explain the decisions of RL agents by determining which training trajectories influenced the actions at a particular state. The proposed method clusters the trajectories in the training data, runs RL on all subsets of $N-1$ clusters, and then uses the differences between the learned policies to determine which clusters affect the decisions at a particular state.

Overall, I think the idea of using attribution to explain RL decisions makes a lot of sense, and is arguably more useful and less subjective than prior approaches based on saliency maps. The paper does a good job explaining the idea (given the 4 page limit). I had a few questions about the method, listed below. My main reservations about the paper are that (1) it's unclear how the method would provide *precise* attributions (without using a huge number of clusters) and (2) the experimental results don't convince me that the method actually works. Concern #2 could be addressed by adding a paragraph explaining Figure 2, and perhaps adding another visualization of the results. To make space for this, the related work section on sequence modeling could be moved to the appendix.

Questions/comments
* The intro says that explainability is important without really providing an argument for why. It'd be good to add some citations to this, or strengthen the argument a bit (e.g., walk through an example scenario where explainable RL methods would be preferred). E.g., why do explanations "confer faith"?
* "of the deep" --> "of deep"
* "requires - i)" --> "requires i)"
* "There is ..." --> I'd recommend cutting this sentence and adding the citations to the previous sentence.
* "experiences(trajectories" --> "experiences (trajectories" (missing space)
* "agent learn certain" -- Grammar
* "behaviors A.6" --> "behaviors (see Appendix A.6)"
* Fig 1 -- Add more details to the caption to explain the method. I suspect that it'd be possible to clarify this figure so that it shows $N$ policies being trained on $N$ subsets of the data (e.g., show a TSNE plot of the trajectories, draw overlapping circles to show which policies are trained on which subsets). Also, figures should appear on the page that they are referenced (if possible).
* Trajectory encoding -- How is this trained? (i.e., what is the objective function?)
* Data embedding -- Why is this step necessary? Why can't we just use the clusters from step (ii)?
* Fig 2 -- Add more details to the caption to explain what the figure is showing. E.g., are these all the trajectories from the attributed cluster?

---

### Official Review · Reviewer_hapK · 2022-10-19
**Interesting and novel idea for explainability in RL**

**Rating:** 7
**Confidence:** 3

**Review:**

The authors propose an attribution-based explainability framework for offline RL, where an agent's decisions are explained by referencing the trajectories in the dataset that produced those decisions. They build upon trajectory/decision transformers to produce trajectory embeddings, cluster these embeddings, and analyze the agent's sensitivity to different clusters of trajectories.

Pros:
- This attribution-based explainability setup is novel with respect to offline RL
- The algorithm is straightforward
- The empirical results are promising
Cons:
- The method seems like it may be intractable to scale to larger offline datasets with a wider variety of behaviors (larger number of clusters)

Overall, the paper is clearly written and well-presented. The algorithm is straightforward and easy to follow. The idea of attribution-based explainability in RL is appealing.